# Metabolic Diseases and Risk of Head and Neck Cancer: A Cohort Study Analyzing Nationwide Population-Based Data

**DOI:** 10.3390/cancers14133277

**Published:** 2022-07-04

**Authors:** Soo-Young Choi, Hyeon-Kyoung Cheong, Min-Kyeong Lee, Jeong-Wook Kang, Young-Chan Lee, In-Hwan Oh, Young-Gyu Eun

**Affiliations:** 1Department of Otolaryngology Head & Neck Surgery, School of Medicine, Kyung Hee University, Seoul 02447, Korea; soo904@naver.com (S.-Y.C.); simbody@naver.com (J.-W.K.); medchan@hanmail.net (Y.-C.L.); 2Department of Internal Medicine, School of Medicine, Korea University, Ansan 15355, Korea; chongmy99@hanmail.net; 3Department of Biomedical Science and Technology, Graduate School, Kyung Hee University, Seoul 02447, Korea; msdbjar@naver.com; 4Department of Preventive Medicine, School of Medicine, Kyung Hee University, Seoul 02447, Korea

**Keywords:** metabolic syndrome, obesity, diabetes, hypertension, head and neck cancer

## Abstract

**Simple Summary:**

Head and neck cancer has been increasing in recent years, and it seriously deteriorates the quality of life because it directly affects the organs of eating, talking and breathing. Therefore, studies to prevent and treat it are being actively conducted. In the present study, we investigated the effects of metabolic diseases on the development of head and neck cancer as part of research to identify risk factors for head and neck cancer and to prevent it. Obesity lowered the risk of HNC in men. Underweight was a strong risk factor for HNC in both men and women. High LDL cholesterol and TC levels had protective effects on HNC in men, but not in women. Diabetes increased the risk of HNC in both men and women, whereas hypertension increased the risk of HNC only in men.

**Abstract:**

The aim of the study was to investigate the association between metabolic diseases and the risk of head and neck cancer (HNC) using nationwide population-based big data. This retrospective cohort study was conducted using the Korean National Health Insurance Service health checkup database. A total of 4,575,818 participants aged >40 years who received a health checkup in 2008 were enrolled, and we studied the incidence of HNC until 2019. We analyzed the risk of HNC according to the presence of metabolic diseases, such as obesity, dyslipidemia, hypertension, and diabetes. Although metabolic syndrome itself was not associated with HNC, each component of metabolic syndrome was associated with HNC. Underweight and diabetes were risk factors for HNC (HR: 1.694). High total cholesterol and high low-density lipoprotein cholesterol levels were factors that decreased the risk (HR 0.910 and 0.839). When we analyzed men and women separately, low total cholesterol level, low low-density lipoprotein cholesterol level, and hypertension were risk factors only in men. In addition, pre-obesity, obesity, and central obesity decreased the risk only in men. Each metabolic disease affects HNC in different ways. Underweight and diabetes increased the risk of HNC, whereas high total cholesterol and high low-density lipoprotein cholesterol levels decreased the risk of HNC.

## 1. Introduction

Worldwide, in 2017, more than 800,000 cases were newly diagnosed as head and neck cancers (HNC), and approximately 500,000 people died due to HNC [1]. HNC refers to cancers of the oral cavity, oropharynx, hypopharynx, and larynx [2], and these areas are associated with eating, talking, and breathing. For these reasons, the onset of HNC affects the quality of life more than that of other cancers, and the treatment method is difficult. Therefore, as with other cancers, it is important to identify and prevent the risk factors that cause HNC. Cigarettes, alcohol, and human papilloma virus are known to be the representative causes of HNC [3].

In recent years, many studies have reported the relationship between various cancers and metabolic diseases [4], Metabolic diseases usually include diabetes, hypertension, obesity, and dyslipidemia. The criteria for the clinical diagnosis of these have been proposed slightly differently by several expert groups [5,6,7,8,9], Diabetes and hypertension have been reported to be associated with various cancers [10,11,12]. Abdominal obesity is a risk factor for cancer and one of the factors that worsens the prognosis [13], and obesity increases the risk of breast cancer [14]. In addition, low high-density lipoprotein (HDL) cholesterol and high triglyceride (TG) levels are associated with lung cancer [15]. Chemoradiotherapy treatment, on the other hand, also exposes the patient to innumerable immune, nutritional and metabolic complications, often affecting the patient’s quality of life [16]. However, the association between the incidence of HNC and these metabolic diseases has not yet been clearly determined. Unlike other cancers, obesity has been reported to have a positive effect on the prognosis of HNC [17]. Since various cancers have different characteristics, in this study, we sought to investigate the association between metabolic diseases and the risk of HNC.

## 2. Materials and Methods

### 2.1. Study Population

The Korean government operates the public health insurance system through an organization called the Korean National Health Insurance Service (KNHIS), and for 97% of the entire population of Korea is compulsory to join [18]. The KNHIS recommends that adults aged > 40 years and employees aged > 20 years receive regular health checkups once every 1–2 years. The KNHIS has data such as demographics, diagnoses, and treatment records of individual diseases along with the health checkup data of subscribers. The health checkup data include information on height, weight, blood test results, and smoking and alcohol drinking status. We used this database from the KNHIS with official approval from the government.

This study was conducted with patients aged > 40 years who underwent a health checkup from the KNHIS in 2008. Only those who had not previously been diagnosed with HNC were included, and we examined whether they had HNC from 2009 to 2019. In 2008, 6,093,826 people underwent routine health checkups. Of these, 1,502,198 were excluded as missing data. A total of 11,893 people were excluded because they were previously diagnosed with HNC. A total of 687 people who died or were diagnosed with HNC within 1 year were also excluded. Lastly, 3230 people who were diagnosed with cancer of the nasal cavity, middle ear, salivary gland, and nasopharynx were excluded. The flow diagram of the study population is demonstrated in Figure 1.

The study was approved by the Institutional Review Board (IRB) of Kyung Hee University Hospital on 8 November, 2019 and was performed in accordance with relevant guidelines and regulations (IRB No. 2019-11-002).

### 2.2. Definition of Variables

Metabolic syndrome was defined according to the International Diabetes Federation criteria [19]. According to this institution, for a person to be defined as having metabolic syndrome, they must have central obesity (waist circumference ≥ 90 cm for men and ≥80 cm for women), plus any two of the following four factors: elevated TG level (≥150 mg/dL or specific treatment for this lipid abnormality), reduced HDL cholesterol level (<40 mg/dL for men and <50 mg/dL for women), elevated blood pressure (systolic ≥ 130 mmHg or diastolic ≥ 85 mmHg or treatment of previously diagnosed hypertension), and elevated fasting plasma glucose level (≥100 mg/dL or previously diagnosed type 2 diabetes) [19]. Body mass index (BMI) was calculated by dividing the weight in kilograms by the square of height in meters (kg/m^2^). Based on BMI, the group was defined as underweight (<18.5 kg/m^2^), normal (18.5–22.9 kg/m^2^), pre-obese (23.0–24.9 kg/m^2^), and obese (≥25 kg/m^2^) [20]. For the lipid profile, when the total cholesterol (TC) level was ≥200 mg/dL, TG level was ≥150 mg/dL, and low-density lipoprotein (LDL) cholesterol level was ≥100 mg/dL, it was classified as high or abnormal, respectively. HDL cholesterol level was defined as low or abnormal when <40 mg/dL in men and <50 mg/dL in women [21]. Hypertension and diabetes were defined as previously diagnosed or treated using the KNHIS data, respectively. Standardized self-reported questionnaires were used to collect data on alcohol-drinking and smoking status at the time of enrollment. Alcohol drinking was classified as nondrinker, moderate drinker (ethanol, 0–30 g/day), and heavy drinker (ethanol, >30 g/day). Smoking status was classified as never smoker, ex-smoker, or current smoker. Age was defined as the age at which the patients were enrolled.

### 2.3. Statistical Analysis

Categorical variables such as sex, smoking and alcohol-drinking status, and past medical history are presented as frequency and percentage. The values of metabolic syndrome are presented as the mean and standard deviation. The differences between the HNC and control groups were analyzed using the chi-square and *t*-test. A Cox proportional hazard regression model was used to determine the relationship between metabolic syndrome and HNC. The hazard ratio and 95% confidence interval (CI) are presented. To determine the effect of sex, sex was stratified. SAS 9.4 (SAS Institute Inc., Cary, NC, USA) was used, and the significance of statistical tests was determined using the 0.05 level.

## 3. Results

### 3.1. Baseline Study Participant Characteristics

The demographic characteristics are summarized in Table 1. A total of 4,575,818 patients were analyzed, and 8749 patients were newly diagnosed with HNC during the analysis period. In the HNC group, 83.53% were men, which was higher than the 53.97% of the non-HNC group (*p* < 0.001). The mean age of the HNC group was 59.53 years, which was higher than that of the non-HNC group (53.96 years) (*p* < 0.001). Among the HNC patients, 5.86% were heavy drinkers, 48.01% were mild drinkers, and 46.13% were nondrinkers, whereas in the non-HNC group, 3.11% were heavy drinkers, 40.66% were mild drinkers, and 56.04% were nondrinkers (*p* < 0.001). The percentages of current smokers (34.82% vs. 20.26%) and ex-smokers (14.58% vs. 10.33%) were higher in the HNC group, while the percentage of never smokers was low (50.60% vs. 69.22%) (*p* < 0.001). The mean BMI was different in the two groups (23.70 kg/m^2^ in the HNC group and 23.98 kg/m^2^ in the non-HNC group) (*p* < 0.001). The mean waist circumferences were 83.78 cm in the HNC group and 81.68 cm in the non-HNC group (*p* < 0.001). Each mean of lipid profiles showed a significant difference in the two groups: the total cholesterol levels were 193.35 md/dL in the HNC group and 197.45 md/dL in the non-HNC group, the TG levels were 146.24 md/dL in the HNC group and 139.80 md/dL in the non-HNC group, the LDL cholesterol levels were 113.56 md/dL in the HNC group and 119.64 md/dL in the non-HNC group, the HDL cholesterol levels were 54.71 md/dL in the HNC group and 55.55 md/dL in the non-HNC group (*p* < 0.001, respectively) The proportions of hypertension were 58.37% in the HNC group and 48.89% in the non-HNC group (*p* < 0.001). The proportions of patients with diabetes were 17.02% in the HNC group and 11.75% in the non-HNC group (*p* < 0.001). In the HNC group, 29.03% met the criteria for metabolic syndrome, whereas in the non-HNC group, 26.16% met the criteria for metabolic syndrome (*p* < 0.001).

### 3.2. Impact of Each Factor on the Risk of HNC

Each factor was subdivided, and the effect of each factor on the risk of HNC was analyzed. When age was subdivided into 40 s to 80 s, the risk of HNC increased with increasing age. The hazard ratios of 40 s, 50 s, 60 s, 70 s, and 80 s were 1.982 (95% CI, 1.862–2.110), 3.000 (95% CI, 2.812–3.201), 3.736 (95% CI, 3.477–4.016), and 5.276 (95% CI, 4.430–6.283), respectively. Multivariable analysis was adjusted for factors including age and sex. Of 8749 patients with HNC, 2540 met the criteria for metabolic syndrome, and 6209 did not, and metabolic syndrome did not affect the adjusted hazard ratio (aHR) for the development of HNC. The aHR of HNC in heavy drinkers (1.498 [95% CI, 1.363–1.646]) was significantly higher than that in moderate drinkers (1.079 (95% CI, 1.030–1.131)) and nondrinkers. In the case of smoking status, the aHR of current smokers (1.668 (95% CI, 1.586–1.754)) was much higher than that of ex-smokers (1.668 (95% CI, 1.586–1.754)). The underweight group (BMI < 18.5 kg/m^2^) was a notable risk factor for HNC (aHR, 1.694 (95% CI, 1.505–1.906)). Inversely, the pre-obese (23.0–24.9 kg/m^2^) and obese (≥25 kg/m^2^) groups had lower aHRs of 0.838 (95% CI, 0.794–0.883) and 0.804 (95% CI, 0.765–0.846), respectively. Similarly, the aHR decreased when patients had central obesity (0.943 (95% CI, 0.879–0.992)). High TC and high LDL cholesterol levels decreased the aHR to 0.910 (95% CI, 0.871–0.949) and 0.839 (95% CI, 0.803–0.876), respectively. However, the TG and HDL cholesterol levels were not risk factors for HNC. When patients had diabetes, the aHR increased (1.181 (95% CI, 1.116–1.249)). Hypertension did not influence the incidence of HNC (Table 2).

### 3.3. Analysis of the HNC Risk by Sex According to Each Factor

The effect of each factor on the risk of HNC was analyzed by subgrouping men and women. Multivariable analysis was adjusted for factors including age and smoking and alcohol drinking status. The underweight group had a significantly increased aHR in both men and women (1.710 (95% CI, 1.506–1.942) and 1.578 (95% CI, 1.155–2.156), respectively). However, the pre-obese and obese groups had a decreased aHR only in men (0.804 (95% CI, 0.759–0.852) and 0.778 (95% CI, 0.736–0.822)). That is, a high BMI did not influence HNC in women. Central obesity also affected HNC only in men. For central obesity, the aHR decreased to 0.937 (95% CI, 0.889–0.988) in men. Similarly, high TC and high LDL cholesterol levels were the factors that decreased the aHR to 0.918 (95% CI, 0.876–0.962) and 0.830 (95% CI, 0.792–0.870) only in men. The TG and HDL cholesterol levels were not associated with HNC in either men or women. Although hypertension had no relationship with HNC when men and women were considered together, it increased the aHR to 1.049 (95% CI, 1.001–1.101) only in the male group when men and women were considered separately. Both subgroups showed significance regarding the risk of diabetes. The aHRs of diabetes were 1.183 (95% CI, 1.114–1.256) and 1.190 (95% CI, 1.016–1.939) in men and women, respectively (Table 3).

## 4. Discussion

Metabolic syndrome is diagnosed with metabolic diseases such as central obesity, glucose intolerance, hypertension, and dyslipidemia. Metabolic syndrome is increasing in various countries, including developed countries, as diabetes and obesity are increasing [22,23]. It is well known that metabolic syndrome affects various diseases, but recent reports have shown that metabolic syndrome increases the risk of various cancers. Previous studies have reported that metabolic syndrome increases the risk of developing and recurring liver, colorectal, bladder, endometrial, pancreatic, and breast cancers [24,25,26,27,28]. However, there have been few studies on the relationship between metabolic syndrome and head and neck cancer. Thus, this study aimed to investigate the association between metabolic syndrome and HNC. However, in our study, metabolic syndrome did not increase the risk of HNC, as in other cancers. This is consistent with a recent cohort study that reported no correlation between metabolic syndrome and head and neck cancer [29]. Our study results show that the components of metabolic syndrome had different effects on HNC. Some components had positive effects, while others had inverse effects. Therefore, we investigated the correlation between each component of metabolic syndrome and the risk of HNC.

This study investigated the effects of metabolic diseases, such as dyslipidemia, obesity, hypertension, and diabetes, on HNC using big data from the KNHIS. In general, obesity and dyslipidemia are risk factors for many diseases, but these are not associated with HNC. In high BMI or central obesity, the risk of HNC decreased. Low TC and low LDL cholesterol levels were also risk factors for HNC. However, these only affected men and not women. Diabetes was a risk factor for HNC in both men and women, and hypertension was a risk factor only in men. In addition, as already known, we confirmed that age, smoking, and alcohol drinking were significant risk factors for HNC.

There have been many studies on the association between lipids and various cancers. Previous studies have shown different views. According to one study, high cholesterol levels increased the risk of prostate cancer, and cholesterol-lowering drugs, such as statins, help prevent prostate cancer [30]. Another study also reported that the use of statins reduced the risk of lung, colorectal, prostate, and bladder cancers and melanoma, suggesting that high cholesterol levels increase the risk of various cancers [31]. They suggested that cholesterol affects the production and inhibition of cancer-related molecules. A similar study reported that high TG levels increased the risk of colorectal cancer, explaining that TG attenuates insulin resistance [25]. A meta-analysis reported that HDL cholesterol and TC had an inverse correlation with lung cancer but TG had a positive correlation. They suggested that this was because lipid metabolism may act as a special mechanism in the etiology of lung cancer [15]. As such, there is disagreement as to whether the blood lipid level is a risk factor for various cancers, and the exact mechanism of how lipids affect the development of cancer is not yet known. However, since lipids are directly or indirectly involved in cell cycle-related metabolism, such as apoptosis and inflammation [32,33,34,35], they may also affect the pathogenesis of cancer. In our study, low TC and LDL cholesterol levels increased the risk of HNC, and the TG and HDL cholesterol levels did not affect the incidence of HNC. Further research is needed to determine whether this is due to the specialty of HNC, various confounding factors, biological differences between men and women, or differences in lifestyles such as smoking and drinking between men and women. Reverse causation was also considered. A study suggested that patients with HNC have poor nutritional intake due to dysphagia and loss of appetite [36]. HNC is much more common in men, and it is possible that the blood lipid levels were lowered due to HNC rather than the HNC risk increased due to low lipid levels.

Obesity is known to be closely related to the metabolism of glucose and lipids and has a great influence on hormones [37]. It has been reported that obesity is associated with various cancers. An extensive study showed that obesity increases the risk of various cancers such as colorectal, breast, endometrial, kidney, esophageal, pancreatic, and gastric cancers. This study shows that obesity destroys the balance of hormones and causes metabolic disorders, which increases the risk of cancer [13]. However, the mechanism by which obesity increases the risk of cancer remains unclear. There are still many disagreements regarding the effect of obesity on the development of HNC. Several cohort studies have reported that HNC is not associated with obesity [36,38] and several studies have reported that a high BMI lowers the incidence of HNC [39,40,41]. Other studies have reported that abdominal obesity increases the risk of HNC [40,42]. It has also been reported that lean people have a poor prognosis for HNC. This is because obesity acts as a buffer in the treatment of HNC, which has protective effects against cachexia, malnourishment, and immune compromise caused by dysphagia and poor appetite, which slows the growth of HNC and inhibits metastasis [17]. If obesity itself makes HNC grow slowly and inhibit metastasis, it may be suggested that obesity has the effect of preventing HNC.

In our study, abdominal obesity and high BMI lowered the risk of HNC only in men. This result has not been presented in previous studies. As mentioned earlier, the reasons for this result have not been demonstrated, but possible reasons can be considered by examining previous studies. First, as seen in our study, men smoke and drink more than women, and many carcinogens from cigarettes damage DNA [43]. It is possible that this molecular mechanism is related to body weight. Lean people have more DNA damage from smoking, which has a greater impact on the development of HNC [44]. Conversely, smoking causes hormonal changes and dietary habits, destroys muscles, and causes low weight [45,46]. Therefore, men smoke much more than women, and because of this, the number of lean people also increases, thereby increasing the incidence of HNC. Second, the incidence of HNC is high because of the high smoking rate among men. Before the diagnosis of HNC, symptoms such as dysphagia and odynophagia could cause low weight. Therefore, it can be seen that low weight increases the incidence of HNC as a confounding effect [47].

Many studies have shown that diabetes increases the incidence of almost all cancers. It seems obvious that diabetes is a risk factor for many types of cancer. A previous study explained that diabetes increases cancer risk through hyperinsulinemia, hyperglycemia, and chronic inflammation. This study demonstrated the mechanisms by which these factors influence cancer and suggested that the control of diabetes helps inhibit the development of cancer [10]. Another cohort study reported that diabetes increases the risk of HNC [48]. This is consistent with our findings. However, another large study reported that diabetes and metabolic syndrome each lowered the risk of head and neck cancer. Further molecular studies, as well as well-controlled demographic studies are needed to elucidate the relationship between diabetes and head and neck cancer [49].

In the case of hypertension, there is controversy. A cohort study conducted in various countries reported a correlation between hypertension and the risk of various cancers. They showed that hypertension also increased cancer mortality [12]. Other studies have suggested a correlation between hypertension and cancer, but some studies have suggested that there is no correlation [50,51]. In addition, to our knowledge, no study has demonstrated a clear mechanism of the effect of hypertension on various cancers. In our study, hypertension was a risk factor for HNC only in men. Smoking and alcohol consumption are risk factors for both hypertension and HNC; therefore, our finding may be due to the high smoking rate and high alcohol consumption in men.

Recent studies have shown that a large number of patients with head and neck cancer have comorbidities and complications. The most common comorbidities were hypertension, metastatic cancer, and lung disease, and the most common complications included hemorrhage/hematoma and postoperative pulmonary function decline. Since the rate of head and neck cancer in the elderly is high, the management of comorbidities and complications will be important [52].

Our study is meaningful as it is the first cohort study of the relationship between HNC and metabolic diseases using nationwide big data. In addition, this study was conducted by subgrouping men and women, and it was possible to closely examine the effects of metabolic diseases on HNC. This can help identify the unique characteristics of HNC that are different from those of other cancers.

## 5. Conclusions

Metabolic syndrome itself had no effect on the incidence of HNC, but a different finding was noted when each metabolic disease constituting metabolic syndrome was studied separately. Obesity, which is generally evaluated as BMI and central obesity, lowered the risk of HNC in men. Underweight was a strong risk factor for HNC in both men and women. High LDL cholesterol and TC levels had protective effects on HNC in men, but not in women. Diabetes increased the risk of HNC in both men and women, whereas hypertension increased the risk of HNC only in men. Further studies are needed to evaluate the effects of metabolic diseases on HNC.

## Figures and Tables

**Figure 1 cancers-14-03277-f001:**
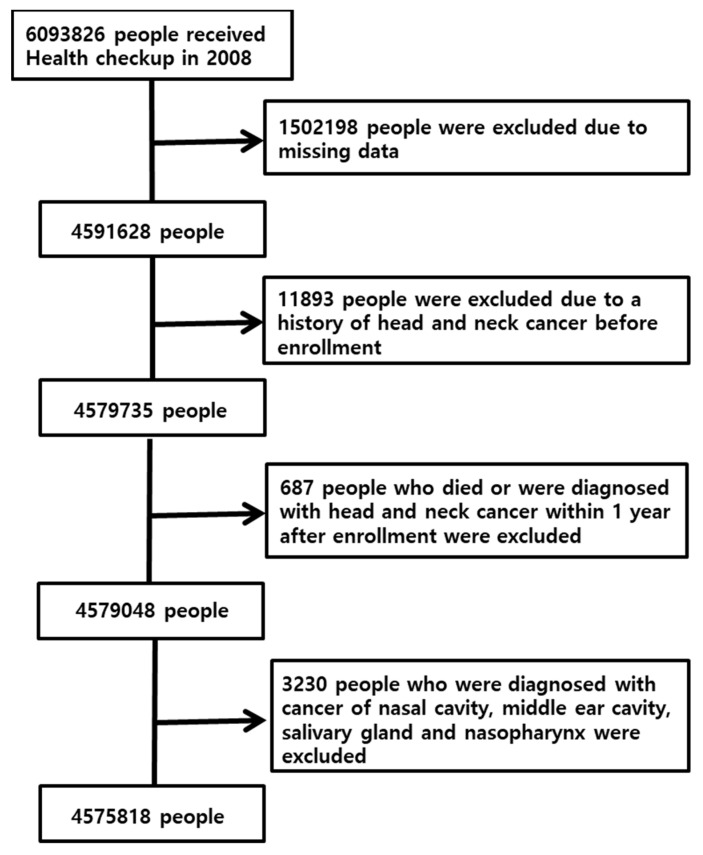
Flow diagram of the study population.

**Table 1 cancers-14-03277-t001:** Analysis of factors potentially associated with head and neck cancer.

Parameter	With HNC (*n* = 8749)	Without HNC (*n* = 4,567,069)	*p* Value
Sex, *n* (%)			<0.001 *
Male	7308 (83.53)	2,469,600 (53,97)	
Female	1441 (16,47)	2,097,469 (45,84)	
Age, year	59.53 ± 9.69	53.96 ± 9.32	<0.001 *
Alcohol drinking, *n* (%)			<0.001 *
No	4036 (46.13)	2,564,134 (56.04)	
Moderate	4200 (48.01)	1,860,466 (40.66)	
Heavy	513 (5.86)	142,469 (3.11)	
Smoking status, *n* (%)			<0.001 *
Never smoker	4427 (50.60)	3,167,225 (69.22)0	
Ex-smoker	1276 (14.58)	472,603 (10.33)	
Current smoker	3046 (34.82)	927,241 (20.26)	
Body mass index, kg/m^2^	23.70 ± 3.00	23.98 ± 3.32	<0.001 *
Waist circumference, cm	83.78 ± 8.16	81.68 ± 8.45	<0.001 *
Total cholesterol, mg/dL	193.35 ± 37.43	197.45 ± 36.95	<0.001 *
Triglyceride, mg/dL	146.24 ± 98.10	197.45 ± 36.95	<0.001 *
LDL cholesterol, mg/dL	113.56 ± 115.86	119.64 ± 79.48	<0.001 *
HDL cholesterol, mg/dL	54.71 ± 31.45	55.55 ± 31.89	0.0144 *
Hypertension, *n* (%)			<0.001 *
No	3642 (41.63)	2,330,154 (50.92)	
Yes	5107 (58.37)	2,236,915 (48.89)	
Diabetes, *n*			<0.001 *
No	7260 (82.98)	4,029,420 (88.06)	
Yes	1489 (17.02)	537,649 (11.75)	
Metabolic syndrome, *n* (%)			<0.001 *
No	6209 (70.97)	3,372,434 (73.84)	
Yes	5540 (29.03)	1,194,635 (26.16)	

Values are mean ± standard deviation (SD) or % ± SD. * Significant at *p* < 0.05.

**Table 2 cancers-14-03277-t002:** Hazard ratio of head and neck cancer risk according to each factor.

Parameter	*n*	HNC (Adjusted for Age and Sex)	
		*p* Value	HR (95% CI)
Alcohol drinking, *n*			
No	4036		1 (reference)
Moderate	4200	0.0013	1.079 (1.030–1.131)
Heavy	513	<0.001	1.498 (1.363–1.646)
Smoking status, *n*			
Never smoker	4427		1 (reference)
Ex-smoker	1276	<0.001	1.185 (1.110–1.265)
Current smoker	3046	<0.001	1.668 (1.586–1.754)
BMI, kg/m^2^			
<18.5	303	<0.001	1.694 (1.505–1.906)
18.5–22.9	3250		1 (reference)
23–24.9	2344	<0.001	0.838 (0.794–0.883)
≥25	2852	<0.001	0.804 (0.765–0.846)
Central obesity			
No	6691		1 (reference)
Yes	2058	0.0218	0.943 (0.897–0.992)
Total cholesterol			
Low	5198		1 (reference)
High	3551	<0.001	0.910 (0.871–0.949)
LDL cholesterol			
Low	3257		1 (reference)
High	5492	<0.001	0.839 (0.803–0.876)
TG			
Low	5657		1 (reference)
High	3092	0.649	1.010 (0.967–1.056)
HDL cholesterol			
Low	7158	0.7573	1.009 (0.955–1.066)
High	1591		1 (reference)
Hypertension			
No	3642		1 (reference)
Yes	5107	0.1977	1.029 (0.985–1.075)
Diabetes			
No	7260		1 (reference)
Yes	1489	<0.001	1.181 (1.116–1.249)
Metabolic syndrome			
No	6209		1 (reference)
Yes	2540	0.3578	1.022 (0.976–1.071)

**Table 3 cancers-14-03277-t003:** Hazard ratio of head and neck cancer risk by sex according to each factor.

Parameter	*n*	Male (*n* = 7308)		*n*	Female (*n* = 1441)	
		*p* Value	HR (95% CI)		*p* Value	HR (95% CI)
BMI, kg/m^2^						
<18.5	260	<0001	1.710 (1.506–1.942)	43	0.0042	1.578 (1.155–2.156)
18.5–22.9	2751	<	1 (reference)	499		1 (reference)
23–24.9	1951	<0001		506	0.1546	1.101 (0.964–1.258)
≥25	2346	<0001		393	0.4324	1.052 (0.928–1.192)
Central obesity						
No	5435		1 (reference)	1256		1 (reference)
Yes	1873	0.016	0.937 (0.889–0.988)	185	0.684	1.033 (0.883–1.208)
Total cholesterol						
Low	4458		1 (reference)	740		1 (reference)
High	2850	0.0004	0.918 (0.876–0.962)	701	0.2519	0.941 (0.847–1.044)
LDL cholesterol						
Low	2862		1 (reference)	395		1 (reference)
High	4446	<0001	0.830 (0.792–0.870)	1046	0.2588	0.935 (0.833–1.05)
TG						
Low	4622		1 (reference)	1035		1 (reference)
High	2686	0.228	1.03 (0.982–1.08)	406	0.8323	1.013 (0.901–1.138)
HDL cholesterol						
Low	6201		1 (reference)	957		1 (reference)
High	1107	0.8254	1.007 (0.945–1.074)	484	0.6007	0.971 (0.87–1.084)
Hypertension						
No	2942		1 (reference)	700		1 (reference)
Yes	4366	0.0467	1.049 (1.001–1.101)	741	0.6007	0.971 (0.87–1.084)
Diabetes						
No	6001		1 (reference)	1259		1 (reference)
Yes	1307	<0001	1.183 (1.114–1.256)	182	0.0307	1.190 (1.016–1.393)

## Data Availability

The data presented in this study are available on request from the corresponding author. The data are not publicly available owing to privacy and ethical reasons.

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
