# Peer review of "Metabolic Diseases and Risk of Head and Neck Cancer: A Cohort Study Analyzing Nationwide Population-Based Data"

_cancers, 2022, doi:10.3390/cancers14133277_

Round 1

Reviewer 1 Report

This very preliminary report by Choi et al. shows the risk of HNC according to the presence of metabolic diseases, such as obesity, dyslipidemia, hypertension, and diabetes. The current article resulted in the identification of no effect of Metabolic syndrome on HNC, but a different finding was noted when each metabolic disease constituting metabolic syndrome was studied separately. Although the paper/manuscript was written in a very well manner, the authors failed to build/show strong correlation between metabolic diseases and Head and Neck Cancer. However, the article adds nothing to the current knowledge. No functional studies/demographic study on other groups of population were performed on the role of identified Underweight, High LDL cholesterol and TC, how they can influence the development of HNC in both men and women. Metabolomics is a new, rapidly expanding field of systems biology that has garnered significant interest in biomedical research and cancer biology. Metabolomics provides a “snapshot” in time of all metabolites present in a biological sample and much advanced information is already available in the context of HNC, author need to incorporate the recent trend in such a type of study. In my opinion it lacks the data and supporting material and needs to be present in another form with additional data and supporting material. 

Reviewer 2 Report

Some minor concernings:

  • after '' associated with lung cancer. [15] '' chemoradiotherapy treatment, on the other hand, also exposes the patient to innumerable immune, nutritional and metabolic complications, often affecting the patient's quality of life. please cite ''doi:10.1016/j.anl.2021.05.007.''

Methods

  • why the study was performed till to 2019?
  • in study population cite table with demographic data
  • consort model for the flow diagram would be interesting
  • ''according to this institution''.. please cite a guidelines

Discussion

An interesting paper analyzed a largest number of head and neck cancer cases involving comorbidities (90.54%) and the highest rate of overall complications(27.50%) occurred in moderate case volume institutions compared to athe complication rate of 22.89% in low volume hospitals and 21.50% in high volume hospitals (P < .0001). The most common comorbidities across all 3 hospital case volumes included hypertension, metastatic cancer,and chronic pulmonary disease and the most common complicationsincluded hemorrhage/hematoma and postoperative pulmonarycompromise. With more geriatric patients requiring surgery for head andneck cancer, it would be beneficial to manage the more complex cases at high volume centers and to develop multidisciplinary teams to optimizecase management and minimize complications.

please cite doi: 10.1177/0145561319856006. 
